# Impacts of Small-Scale Effect and Nonlinear Damping on the Nonlinear Vibrations of Electrostatic Microresonators

**DOI:** 10.3390/mi14010170

**Published:** 2023-01-09

**Authors:** Dayao Meng, Kun Huang, Wei Xu

**Affiliations:** Department of Engineering Mechanics, Faculty of Civil Engineering and Mechanics, Kunming University of Science and Technology, Kunming 650500, China

**Keywords:** microbeam resonators, electric actuation, scale effect, nonlinear vibration, primary resonance, multiple-scale method

## Abstract

Although the small-scale effect and nonlinear damping on the nonlinear vibration of microbeam electrostatic resonators are important, they have been overlooked by researchers. We use the slender beam model including the small-scale effect and nonlinear damping to investigate the nonlinear vibrations of the electrostatic resonators in the present paper. We apply the Galerkin method on a nonlinear partial differential equation to obtain the nonlinear ordinary differential equations for the first and third modes. The two equations include constant terms. The multiple-scale method is used to obtain the approximate analytical solutions of the two equations. The approximate analytical solutions discover the effects of driving electric field, small-scale effect, and nonlinear damping on structural vibrations. The results suggest that the small-scale effect, the direct current (DC) voltage, and the alternating current (AC) voltage have some critical effects on the vibrations of microresonators.

## 1. Introduction

Currently, microresonators are widely used in micro-electro-mechanical systems (MEMS) to perform the sensing and driving functions of MEMS [1]. Given the fast dynamic response, small power consumption and high driving efficiency of electric actuation [2], this study focuses on an electrostatic microresonator. According to the existing results, the large displacement [3], material nonlinearity, damping nonlinearity [4], and scale effect [5] all have noticeable effects on the vibration of microbeam. Therefore, it is necessary to profoundly research these effects on nonlinear vibrations of microbeam electrostatic resonators.

With the advancement of MEMS, there has been more and more attention paid to the impact of complex environments on MEMS dynamics [6]. To optimize the design of these MEMS devices, it is essential to fully understand the statics/dynamics of the system [7,8,9]. Abdel-Rahman applied the multiple-scale method to study the response of a microbeam-based resonant sensor to superharmonic and subharmonic electric actuations. Through discussion about the dynamic bifurcation characteristics of the system, it was found out that there are multiple steady-state solutions for the given parameters [10]. Younis proposed to generate reduced-order models in a different way for electrically actuated microbeam-based MEMS. The model accounts for the nonlinear elastic restoring forces and the nonlinear electric forcing [11]. Hu applied the energy method to develop a mechanical model of a micro-cantilever beam and explored the impact of DC voltage on the static deformation of the micro-cantilever beam [12]. Najar performed a simulation to investigate the dynamics and global stability of a beam-based electrostatic micro-actuator [13]. It was revealed that the basin of attraction depends on the amplitude and frequency of the AC voltage. Furthermore, the smoothness of the boundary on the basin of attraction can be lost and replaced by fractal tongues, which increases the sensitivity of the microbeam to the initial conditions significantly. Younis et al. [14,15] adopted the Galerkin method, differential orthogonal method, and target method to study the static pull-in behavior and dynamic pull-in behavior of microbeams with electric actuation. Han suggested research on the dynamic evolution of the primary frequency response from a prebuckling microbeam-based resonator with $Z_{2}$ symmetry. It was shown that primary frequency response can be divided into two parts: low-energy branch and high-energy branch. As AC excitation increases, they get close to each other along the backbone curve [16]. Younesian applied the Galerkin method to construct the single-degree-of-freedom model of clamped–clamped microbeam resonator and the multi-scale method to analyze the primary and secondary resonances of the beam [17]. Li dealt with the design of some MEMS device motions, such as monostable motion, by avoiding the hardening to softening switch of the low-energy frequency response branch. As a result, dynamic bifurcation was eliminated, and the stability of the system was improved [18].

In MEMS, the size of beams can reach the micron level, and the mechanical properties of materials are closely related to the scale. Since small-scale effect is not considered by the classical continuum mechanics [5], the modified strain gradient theory [19] and the modified couple stress theory [20,21] were proposed in some research to capture the scale effect of the continuum constitutive model. When the dynamic response of micro/nanomaterials is studied, the impact of nonlinearity is a significant factor [22,23,24]. However, the nonlinear term caused by damping is ignored in most studies of micro/nanobeam vibration. For example, Kahrobaiyan only considered the linear term of damping [19], and Xia ignored it [25]. The occurrence of energy dissipation is related to the vibration frequency of the structure. Since the vibration frequency of the microbeam is significantly higher compared to the macrobeam, nonlinear damping plays a major role in the vibration of the microbeam [26,27]. Despite the significant impact of small-scale effect and material nonlinearity on the mechanical properties of micro/nanobeams, their combined effects are discounted by researchers. Huang proposed two new nonlinear non-local Euler–Bernoulli theories by considering the material nonlinearity and the small-scale effect to model the mechanical properties of extensible or inextensible nanobeams [28]. In addition, the new model was also used to analyze the static bending and forced vibration of single-walled carbon nanotubes (SWCNTs). The results show that the material nonlinearity and scale effect can have a significant impact on the mechanical properties of SWCNT. Huang proposed a new Bernoulli–Euler theory of microbeams for the consideration of small-scale effects and nonlinear terms [27]. His research shows that nonlinear damping scarcely affects the small-amplitude vibrations if the frequency of the load is greater than the modal frequency, while nonlinear damping can significantly change the bifurcation points of the load and strongly affects the vibrations under the primary resonance condition.

This paper attempts to study the effects of small-scale effect and nonlinear damping on the microbeam resonator. We added a load actuated by the monopolar plate electrode to the model established by Huang [27]. We first use the Galerkin method to discrete the partial differential equation, and then analyze the resonator’s static bending. We will also use the multiple-scale method to solve the forced vibration equation containing constant terms for the primary resonance case. The effects of nonlinear damping, small-scale effect, DC voltage, and AC voltage on the microresonator will be profoundly discussed. This may help people better understand the nonlinear vibration behavior of microresonators and provides some theoretical basis for practical application.

## 2. Methods

Herein, a hinged–hinged slender microbeam is considered. Based on the Euler-Bernoulli displacement hypothesis, Huang established a nonlinear dynamic equation to model the mechanical properties of the microresonator [27], as shown in Figure 1. This model takes into account the small-scale effect and nonlinear terms as induced by the axial elongation and Kelvin-Voigt damping, as shown in Equation (1).
(1)$$\begin{array}{l} m\frac{\partial^{2}\tilde{w}}{\partial {\tilde{t}}^{2}}+\tilde{E}\frac{\partial }{\partial \tilde{t}}\left[I\frac{\partial^{4}\tilde{w}}{\partial {\tilde{x}}^{4}}-\frac{3S}{2}{\left(\frac{\partial \tilde{w}}{\partial \tilde{x}}\right)}^{2}\frac{\partial^{2}\tilde{w}}{\partial {\tilde{x}}^{2}}\right]+P_{0}\frac{\partial^{2}\tilde{w}}{\partial {\tilde{x}}^{2}}\\ +\left(EI+GS\zeta^{2}\right)\frac{\partial^{4}\tilde{w}}{\partial {\tilde{x}}^{4}}-\frac{3ES}{2}{\left(\frac{\partial \tilde{w}}{\partial \tilde{x}}\right)}^{2}\frac{\partial^{2}\tilde{w}}{\partial {\tilde{x}}^{2}}=F. \end{array}$$
where $\tilde{w}$ is the vertical deflection. $G=E/\left[2\left(1+v\right)\right]$, and *E* and *ν* are the Young’s modulus and Poisson’s ration, respectively. $\tilde{E}$ is the viscous damping coefficient, ζ is the material length scale parameter obtained from experiments, $P_{0}$ is the initial axial load, and *m* refers to the mass per unit length. *S* and *I* are the cross-sectional area and moment of inertia, respectively. S=bh and $I=bh^{3}/12$, where *b* and *h* are referred to as the width and thickness of the beam, respectively. From Equation (1), it can be found out that the small-scale effect has a significant impact on the static bending of microbeams [27]. In Ref. [27], the Kelvin–Viogt damping model is used to describe the energy dissipation of microstructures. Although, it is an open question how to describe the microstructure’s energy dissipation. Recent studies have shown that the Kelvin–Viogt damping model is qualitatively suitable for the vibration of microbeams. For example, Ref. [26] shows that nonlinear dissipation has a significant impact on the dynamics of micromechanical systems, and the Voigt–Kelvin viscoelastic constitutive law can obtain consistent results with the experiments.

A uniform parallel electrode lying under the beam is used to excite a generalized electric force $F\left(x,t\right)$. The load comprises a DC component (polarization voltage) $V_{DC}$ and a small AC component $V_{AC}$, and can be expressed as [15]:(2)$$F\left(x,t\right)=\frac{\varepsilon_{0}b{\left[V_{DC}+V_{AC}\cos\left(\tilde{\Omega }t\right)\right]}^{2}}{2{\left(d-\tilde{w}\right)}^{2}}.$$
where $V_{DC}$ is the DC polarization voltage; $V_{AC}$ and $\tilde{\Omega }$ are the amplitude and frequency of the AC voltage, respectively; $\varepsilon_{0}$ is the dielectric constant of the gap medium; *d* is the distance between the beam and the electrode.

In order to better deal with Equation (1), it may be written in the non-dimensional form. This process can be conducted by introducing the following non-dimensional variables:(3)$$x=\frac{\tilde{x}}{l},\;w=\frac{\tilde{w}}{d},\;t=\omega_{0}\tilde{t},\;\Omega =\frac{\tilde{\Omega }}{\omega_{0}},\;\omega_{0}=\sqrt{\frac{EI+GS\zeta^{2}}{ml^{4}}}.$$

By substituting Equation (3) into Equation (1), we obtain
(4)$$\begin{array}{l} \frac{\partial^{2}w}{\partial t^{2}}+\frac{\partial^{4}w}{\partial x^{4}}+P\frac{\partial^{2}w}{\partial x^{2}}+C_{1}\frac{\partial^{5}w}{\partial t\partial x^{4}}-D{\left(\frac{\partial w}{\partial x}\right)}^{2}\frac{\partial^{2}w}{\partial x^{2}}\\ -C_{2}\frac{\partial }{\partial t}\left[{\left(\frac{\partial w}{\partial x}\right)}^{2}\frac{\partial^{2}w}{\partial x^{2}}\right]=Q\frac{{\left[V_{DC}+V_{AC}\cos\left(\Omega t\right)\right]}^{2}}{{\left(1-w\right)}^{2}}. \end{array}$$
where the new parameters in Equation (4) are expressed as
(5)$$P=\frac{P_{0}}{ml^{2}\omega_{0}^{2}},\;C_{1}=\frac{\tilde{E}I}{ml^{4}\omega_{0}},\;C_{2}=\frac{3\tilde{E}Sd^{2}}{2ml^{4}\omega_{0}},\;D=\frac{3ESd^{2}}{2ml^{4}\omega_{0}^{2}},\;Q=\frac{\varepsilon_{0}b}{2md^{3}\omega_{0}^{2}}.$$

For a hinged–hinged beam, the boundary conditions are expressed as
(6)$$w\left(0,t\right)=w\left(1,t\right)=0,\;\frac{\partial^{2}w}{\partial x^{2}}\left(0,t\right)=\frac{\partial^{2}w}{\partial x^{2}}\left(1,t\right)=0.$$

It is difficult to obtain an accurate analytical solution because Equation (4) is a nonlinear differential equation. Therefore, a reduced-order model is constructed by discretizing Equation (4) into a finite-degree-of-freedom system that consists of ordinary differential equations in time [29,30]. Suppose that the solution of Equation (4) can be written as $w=\sum_{j=1}^{\infty }\sin\left(j\pi x\right)u_{j}\left(t\right)$. Since the second mode of Galerkin discrete loses the square terms, only the first and third modes are studied in this paper. Thus, we have
(7)$$w_{1}=\sin\left(\pi x\right)u_{1}\left(t\right),\;w_{2}=\sin\left(3\pi x\right)u_{2}\left(t\right).$$

According to Younis’ research, the neglected higher-order terms will make difficult of the Taylor-series expansion for the electric-force term [11]. Therefore, Equation (7) is substituted into Equation (4) and, multiplied by ${\left(1-w\right)}^{2}\sin\left(n\pi x\right)$ at both sides of the equations and integrated in the interval [0, 1]. If the fourth and fifth higher-order terms are removed from the equation, we have
(8)$$\begin{array}{l} \ddot{u}+\omega_{j}^{2}u=2c_{1j}\dot{u}+c_{2j}\dot{u}u^{2}+c_{3j}\dot{u}u+k_{1j}u^{3}+k_{2j}u^{2}\\ +k_{3j}\ddot{u}u^{2}+k_{4j}\ddot{u}u+F_{1j}+F_{2j}\cos\left(\Omega t\right)+F_{3j}\cos\left(2\Omega t\right),\;j=1,2. \end{array}$$
where j=1 means the first mode and j=2 means the third mode. The parameters of the first mode are expressed as follows.
(9)$$\begin{array}{l} \omega_{1}^{2}=\pi^{4}-P\pi^{2}\;,c_{11}=-\pi^{4}C_{1}/2,\;c_{21}=-\frac{3\pi^{4}\left(C_{1}+C_{2}\right)}{4},\;c_{31}=\frac{16\pi^{3}C_{1}}{3},\\ k_{11}=\frac{3P\pi^{2}}{4}-\frac{D\pi^{4}}{4}-\frac{3}{4}\pi^{4},\;k_{21}=\frac{16}{3}\pi^{3}-\frac{16P\pi }{3},\;k_{31}=-\frac{3}{4},\;k_{41}=\frac{16}{3\pi },\\ F_{11}=\frac{2Q}{\pi }\left(2V_{DC}{}^{2}+V_{AC}{}^{2}\right),\;F_{21}=\frac{8Q}{\pi }V_{DC}V_{AC},\;F_{31}=\frac{2Q}{\pi }V_{AC}{}^{2}. \end{array}$$

The parameters of the third mode are expressed as follows.
(10)$$\begin{array}{l} \omega_{2}^{2}=81\pi^{4}-9P\pi^{2},\;c_{12}=-81\pi^{4}C_{1}/2,\;c_{22}=-\frac{243\pi^{4}\left(C_{1}+C_{2}\right)}{4},\;c_{32}=144\pi^{3}C_{1},\\ k_{12}=\frac{27P\pi^{2}}{4}-\frac{81D\pi^{4}}{4}-\frac{243}{4}\pi^{4},\;k_{22}=144\pi^{3}-16P\pi ,\;k_{32}=-\frac{3}{4},\;k_{42}=\frac{16}{9\pi },\\ F_{12}=\frac{2Q}{3\pi }\left(2V_{DC}{}^{2}+V_{AC}{}^{2}\right),\;F_{22}=\frac{8Q}{3\pi }V_{DC}V_{AC},\;F_{33}=\frac{2Q}{3\pi }V_{AC}{}^{2}. \end{array}$$

From Equations (8)–(10), we can find that although the microbeam’s model comes from Ref. [27], there are two new crucial features induced by the electrical load. One is that some nonlinear terms with scale effects appear in Equation (8). These coupling terms will have a remarkable effect on the structure’s vibrations, as shown in the next section. Second, the constant terms $F_{1j}$ appear in Equation (8). It has not been profoundly studied to solve a nonlinear ordinary differential equation with constant terms. In this paper, we will solve, for the first time, the equations using the multiscale methods. 

Notably, the structure will undergo buckling if $\pi^{4}-P\pi^{2}=0$. Therefore, we restrict $P<\pi^{2}$ and $P_{0}<\pi^{2}ml^{2}\omega_{0}^{2}$ to avoid the beam’s buckling. In this paper, we refer to Li’s article to set the material of the MEMS resonator as an alloy for study [31]. The following physical parameters in Table 1 are used.

For simplicity, the following four models are considered: the scale-dependent model with nonlinear damping (SDND) for ξ≠0, $c_{2}\neq 0$, and $c_{3}\neq 0$; the scale-independent model (SIM) for ξ=0, $c_{2}\neq 0$, and $c_{3}\neq 0$; the nonlinear damping-independent model (NDIM) for ξ≠0, $c_{2}=0$, and $c_{3}=0$; the scale-independent model without nonlinear damping (SIND) for ξ=0, $c_{2}=0$, and $c_{3}=0$.

## 3. Static Response of Microbeam

We can obtain the static equation of the microresonator if $\ddot{u}$ and $\dot{u}$ in Equation (8) are removed, as in the following:(11)$$\omega^{2}u=k_{1}u^{3}+k_{2}u^{2}+F.$$
where
(12)$$F=\frac{4Q{\left[V_{DC}+V_{AC}\right]}^{2}}{\pi }.$$

Here we use $V_{DC}=1.5\;V$ and $V_{AC}=0.08\;V$ to study the impacts of the scale effect and the initial axial load, as shown in Figure 2. The figure shows that the microresonator does not buckle when 0<F<30, and the small-scale effect reduces the deformations of static bending. Conversely, the initial compressed force will increase the deformation. This influence will increase when the beam is subjected to the combined of the scale effect and the initial load.

## 4. The Approximate Solution for the Primary Resonance

The multiple-scale method is used to solve Equation (8). The aim is to find an approximate solution of the equation for small but finite deformations [32]. We suppose
(13)$$u=\varepsilon u_{1}\left(T_{0},T_{1},T_{2}\right)+\varepsilon^{2}u_{2}\left(T_{0},T_{1},T_{2}\right)+\varepsilon^{3}u_{3}\left(T_{0},T_{1},T_{2}\right).$$
where ε=0.1 is a small parameter and $T_{n}=\varepsilon^{n}t$ (n=0,1,2). We perturb the electric-force constant term $F_{1}$ to the linear differential equation at ε, and the excitation term $F_{2}$ and $F_{3}$ at $\varepsilon^{3}$, so let
(14)$$F_{1}=\varepsilon f_{1},\;F_{2}=\varepsilon^{3}f_{2},\;F_{3}=\varepsilon^{3}f_{3}.$$

In order to make the damping terms appear in the same perturbation equations as the nonlinear terms, it is assumed that $c_{1}=\varepsilon^{2}{\bar{c}}_{1}$. A detuning parameter σ is introduced, and it is supposed that $\Omega =\omega_{n}+\varepsilon^{2}\sigma$ [32]. Then Equation (14) is substituted into Equation (8), the superscript of $\bar{c}$ is removed, obtaining
(15)$$\begin{array}{l} \ddot{u}+\omega^{2}u=2\varepsilon^{2}c_{1}\dot{u}+c_{2}\dot{u}u^{2}+c_{3}\dot{u}u+k_{1}u^{3}+k_{2}u^{2}+k_{3}\ddot{u}u^{2}\\ +k_{4}\ddot{u}u+\varepsilon f_{1}+\varepsilon^{3}f_{2}\cos\left(\omega t+\varepsilon^{2}\sigma t\right)+\varepsilon^{3}f_{3}\cos\left(2\omega t+2\varepsilon^{2}\sigma t\right). \end{array}$$

Substituting Equation (13) into Equation (15) and equating coefficients of powers of ε, we obtain that
(16)$$\varepsilon :\;D_{0}{}^{2}u_{1}+\omega^{2}u_{1}=f_{1}.$$
(17)$$\varepsilon^{2}:\;D_{0}{}^{2}u_{2}+\omega^{2}u_{2}=-2D_{0}D_{1}u_{1}+k_{4}u_{1}D_{0}^{2}u_{1}+c_{3}u_{1}D_{0}u_{1}+k_{2}u_{1}^{2}.$$
(18)$$\begin{array}{l} \varepsilon^{3}:\;D_{0}{}^{2}u_{3}+\omega^{2}u_{3}=-2D_{0}D_{1}u_{2}-2D_{0}D_{2}u_{1}-D_{1}^{2}u_{1}+2c_{1}D_{0}u_{1}\\ +k_{1}u_{1}^{3}+2k_{2}u_{1}u_{2}+c_{2}u_{1}^{2}D_{0}u_{1}+c_{3}u_{1}D_{1}u_{1}+c_{3}u_{1}D_{0}u_{2}\\ +c_{3}u_{2}D_{0}u_{1}+k_{3}u_{1}^{2}D_{0}^{2}u_{1}+k_{4}u_{1}D_{0}^{2}u_{2}+k_{4}u_{2}D_{0}^{2}u_{1}+2k_{4}u_{1}D_{0}D_{1}u_{1}\\ +f_{2}\cos\left(\omega T_{0}+\sigma T_{2}\right)+f_{3}\cos\left(2\omega T_{0}+2\sigma T_{2}\right). \end{array}$$
where $D_{n}=\partial /\partial T_{n}$,n=0,1,2 and the differential operator $D_{n}$ indicates the derivative with respect to the timescale $T_{n}$. The general solution of Equation (16) is
(19)$$u_{1}=A_{1}\left(T_{1},T_{2}\right)e^{i\omega T_{0}}+{\bar{A}}_{1}\left(T_{1},T_{2}\right)e^{-i\omega T_{0}}+\frac{f_{1}}{\omega^{2}}.$$

By substituting Equation (19) into Equation (17), we obtain
(20)$$\begin{array}{l} D_{0}{}^{2}u_{2}+\omega^{2}u_{2}=\left(-2i\omega D_{1}A+\frac{2f_{1}k_{2}A}{\omega^{2}}+\frac{if_{1}c_{3}A}{\omega }-f_{1}k_{4}A\right)e^{i\omega T_{0}}\\ +\left(k_{2}A^{2}+ic_{3}\omega A^{2}-k_{4}\omega^{2}A^{2}\right)e^{2i\omega T_{0}}+2k_{2}A\bar{A}-2k_{4}\omega^{2}A\bar{A}+\frac{f_{1}^{2}k_{2}}{\omega^{4}}+cc. \end{array}$$
where *cc* denotes the complex conjugate of the preceding terms. To avoid secular terms in Equation (20), it is supposed that
(21)$$D_{1}A=-\frac{if_{1}k_{2}A}{\omega^{3}}+\frac{f_{1}c_{3}A}{2\omega^{2}}+\frac{if_{1}k_{4}A}{2\omega }.$$

Then we obtain
(22)$$u_{2}=\left(k_{2}+ic_{3}\omega -k_{4}\omega^{2}\right)\frac{-A^{2}e^{2i\omega T_{0}}}{3\omega^{2}}+\left(\frac{k_{2}}{\omega^{2}}-k_{4}\right)2A\bar{A}+\frac{f_{1}^{2}k_{2}}{\omega^{6}}+cc.$$

By substituting Equations (19), (21), and (22) into Equation (18), it can be known that
(23)$$\begin{array}{l} D_{0}{}^{2}u_{3}+\omega^{2}u_{3}=[2i\omega (-A^{\prime }+c_{1}A+\frac{1}{2}c_{2}A^{2}\bar{A})+(3k_{1}+\frac{10k_{2}^{2}}{3\omega^{2}}+\frac{1}{3}c_{3}^{2}-3k_{3}\omega^{2}\\ +\frac{1}{3}k_{4}^{2}\omega^{2}+\frac{ik_{2}c_{3}}{\omega }-\frac{11}{3}k_{2}k_{4})A^{2}\bar{A}+(\frac{ic_{2}f_{1}^{2}}{\omega^{3}}+\frac{3f_{1}^{2}k_{1}}{\omega^{4}}+\frac{3f_{1}^{2}k_{2}^{2}}{\omega^{6}}+\frac{f_{1}^{2}c_{3}^{2}}{4\omega^{4}}-\frac{f_{1}^{2}k_{3}}{\omega^{2}}\\ -\frac{3f_{1}^{2}k_{4}^{2}}{4\omega^{2}}+\frac{if_{1}^{2}k_{2}c_{3}}{\omega^{5}}+\frac{if_{1}^{2}c_{3}k_{4}}{\omega^{3}})A+\frac{1}{2}f_{2}e^{i\sigma T_{2}}]e^{i\omega T_{0}}+cc+NST. \end{array}$$
where the prime denotes the derivatives with respect to $T_{2}$, and *NST* denotes non-secular terms. To avoid secular terms in Equation (23), it is assumed that
(24)$$\begin{array}{l} 2i\omega (-A^{\prime }+c_{1}A+\frac{1}{2}c_{2}A^{2}\bar{A})+(3k_{1}+\frac{10k_{2}^{2}}{3\omega^{2}}+\frac{1}{3}c_{3}^{2}-3k_{3}\omega^{2}+\frac{1}{3}k_{4}^{2}\omega^{2}\\ +\frac{ik_{2}c_{3}}{\omega }-\frac{11}{3}k_{2}k_{4})A^{2}\bar{A}+(\frac{ic_{2}f_{1}^{2}}{\omega^{3}}+\frac{3f_{1}^{2}k_{1}}{\omega^{4}}+\frac{3f_{1}^{2}k_{2}^{2}}{\omega^{6}}+\frac{f_{1}^{2}c_{3}^{2}}{4\omega^{4}}-\frac{f_{1}^{2}k_{3}}{\omega^{2}}\\ -\frac{3f_{1}^{2}k_{4}^{2}}{4\omega^{2}}+\frac{if_{1}^{2}k_{2}c_{3}}{\omega^{5}}+\frac{if_{1}^{2}c_{3}k_{4}}{\omega^{3}})A+\frac{1}{2}f_{2}e^{i\sigma T_{2}}=0. \end{array}$$

We take A in polar form as
(25)$$A\left(T_{1},T_{2}\right)=\frac{1}{2}a(t)e^{i\theta (t)},\;\bar{A}\left(T_{1},T_{2}\right)=\frac{1}{2}a(t)e^{-i\theta (t)},$$
and introduce Equation (25) into Equation (24). By separating the result into real and imaginary components and introducing $\gamma =\theta -\sigma T_{2}$, we obtain that
(26)$$\left\{\begin{array}{l} a^{\prime }=ac_{1}+\frac{a^{3}c_{2}}{8}+\frac{a^{3}k_{2}c_{3}}{8\omega^{2}}+\frac{ac_{2}f_{1}^{2}}{2\omega^{4}}+\frac{af_{1}^{2}k_{2}c_{3}}{2\omega^{6}}+\frac{af_{1}^{2}c_{3}k_{4}}{2\omega^{4}}-\frac{f_{2}\sin\gamma }{2\omega }\\ \gamma^{\prime }=-\frac{3a^{2}k_{1}}{8\omega }-\frac{5a^{2}k_{2}^{2}}{12\omega^{3}}-\frac{a^{2}c_{3}^{2}}{24\omega }+\frac{3}{8}a^{2}k_{3}\omega -\frac{1}{24}a^{2}k_{4}^{2}\omega +\frac{11a^{2}k_{2}k_{4}}{24\omega }\\ -\frac{3f_{1}^{2}k_{1}}{2\omega^{5}}-\frac{3f_{1}^{2}k_{2}^{2}}{2\omega^{7}}-\frac{f_{1}^{2}c_{3}^{2}}{8\omega^{5}}+\frac{f_{1}^{2}k_{3}}{2\omega^{3}}+\frac{3f_{1}^{2}k_{4}^{2}}{8\omega^{3}}-\frac{f_{2}\cos\gamma }{2\omega a}-\sigma \end{array}\right..$$

Therefore, the second approximation is as follows:(27)$$\begin{array}{l} u=\frac{\varepsilon f_{1}}{\omega^{2}}+\varepsilon a\cos(\theta +\omega t)+\frac{\varepsilon^{2}f_{1}^{2}k_{2}}{\omega^{6}}+\frac{\varepsilon^{2}a^{2}k_{2}}{2\omega^{2}}-\frac{1}{2}a^{2}\varepsilon^{2}k_{4}-\frac{a^{2}\varepsilon^{2}k_{2}\cos(2\theta +2\omega t)}{6\omega^{2}}\\ +\frac{a^{2}\varepsilon^{2}c_{3}\sin(2\theta +2\omega t)}{6\omega }+\frac{1}{6}a^{2}\varepsilon^{2}k_{4}\cos(2\theta +2\omega t)+O\left(\varepsilon^{3}\right). \end{array}$$

The steady-state motions occur when $a^{\prime }=\gamma^{\prime }=0$, which corresponds to the singular points of Equation (26). In this case, the vibration amplitude can be obtained from the following equation as
(28)$$\begin{array}{l} \frac{f_{2}^{2}}{4\omega^{2}}=[{\left(c_{1}+\frac{a^{2}c_{2}}{8}+\frac{a^{2}k_{2}c_{3}}{8\omega^{2}}+\frac{c_{2}f_{1}^{2}}{2\omega^{4}}+\frac{f_{1}^{2}k_{2}c_{3}}{2\omega^{6}}+\frac{f_{1}^{2}c_{3}k_{4}}{2\omega^{4}}\right)}^{2}\\ +(-\sigma -\frac{3a^{2}k_{1}}{8\omega }-\frac{5a^{2}k_{2}^{2}}{12\omega^{3}}-\frac{a^{2}c_{3}^{2}}{24\omega }+\frac{3}{8}a^{2}k_{3}\omega -\frac{1}{24}a^{2}k_{4}^{2}\omega \\ +\frac{11a^{2}k_{2}k_{4}}{24\omega }-\frac{3f_{1}^{2}k_{1}}{2\omega^{5}}-\frac{3f_{1}^{2}k_{2}^{2}}{2\omega^{7}}-\frac{f_{1}^{2}c_{3}^{2}}{8\omega^{5}}+\frac{f_{1}^{2}k_{3}}{2\omega^{3}}+\frac{3f_{1}^{2}k_{4}^{2}}{8\omega^{3}}{)}^{2}]a^{2}. \end{array}$$

By assuming a≠0, Equation (28) is rewritten as 


(29)
$$\begin{array}{l} \sigma =\left(-\frac{3k_{1}}{8\omega }\right.-\frac{5k_{2}^{2}}{12\omega^{3}}-\frac{c_{3}^{2}}{24\omega }+\frac{3}{8}k_{3}\omega -\frac{1}{24}k_{4}^{2}\omega +\left.\frac{11k_{2}k_{4}}{24\omega }\right)a^{2}+\left(-\frac{3k_{1}}{2\omega^{5}}\right.-\frac{3k_{2}^{2}}{2\omega^{7}}\\ +-\frac{c_{3}^{2}}{8\omega^{5}}+\frac{k_{3}}{2\omega^{3}}+\left.\frac{3k_{4}^{2}}{8\omega^{3}}\right)f_{1}^{2}\pm \left[\frac{f_{2}^{2}}{4\omega^{2}a^{2}}-\right.{\left.{\left(c_{1}+\frac{a^{2}c_{2}}{8}+\frac{a^{2}k_{2}c_{3}}{8\omega^{2}}+\frac{c_{2}f_{1}^{2}}{2\omega^{4}}+\frac{f_{1}^{2}k_{2}c_{3}}{2\omega^{6}}+\frac{f_{1}^{2}c_{3}k_{4}}{2\omega^{4}}\right)}^{2}\right]}^{\frac{1}{2}}. \end{array}$$


Equations (28) and (29) represent the amplitude of vibration as a function of the electric-force term, taking into account the influence of scale effect and nonlinear damping for primary resonance. These equations can be used to analyze the vibration of microresonators. The amplitude of the third mode is much smaller than that of the first mode, as shown in Figure 3. In this paper, the amplitude of the third mode excited by electric force is excessively low, so that only the first mode is considered.

In order to verify the results of theoretical analysis, the time evolution and phase portraits of primary resonance are calculated using the Runge–Kutta method for $V_{DC}=1.5\;V$ and $V_{AC}=0.08\;V$, as shown in Figure 4 and Figure 5. The multiple-scale method is compared with the Runge–Kutta method, as shown in Figure 6. The figure indicated that the results of theoretical analysis are accurate.

## 5. Impact of Small-Scale Effect, Nonlinear Damping and Driving Electric Field

It can be seen from Equation (28) that the scale effect has a significant effect on the amplitude of vibration, as shown in Figure 7. These effects are shown that the scale effect and the nonlinear damping not only increase the amplitude near the jump point but also cause the shift of jump points.

According to Equation (9), $V_{AC}$ and $V_{DC}$ have similar effects on the vibrational amplitude, so we will discuss them. As shown in Figure 8 and Figure 9, the small-scale effect leads to a significant reduction in the amplitude of vibration. As an example shown in Figure 9, when $c_{1}=0.01$, the jump will occur at $V_{AC}=0.095\;V$ for the SDND, while the jump will occur at $V_{AC}=0.007\;V$ for the SIM. This indicates that the small-scale effect causes the outstanding shift of jump points, and the vibration amplitudes will decrease with the increase of damping coefficient.

According to Equation (9), the damping coefficients $c_{1}$, $c_{2}$, and $c_{3}$ will increase with the viscous damping coefficient $\tilde{E}$. As shown in Figure 10, the nonlinear damping makes little difference to amplitude vibration when the load’s frequency is less than the modal frequency of the microbeam. As shown in Figure 11, given a small external excitation, which means the values of $V_{AC}$ and $V_{DC}$ are small at the jump point, the amplitude-frequency response curves of SDND and NDIM are similar, namely, the nonlinear damping barely affects amplitude vibration.

Figure 12 shows that the impact of nonlinear damping on primary resonance is reflected mainly in two aspects as follows. Firstly, when the frequency of the load exceeds the modal frequency, the nonlinear damping causes the left bias. Greater nonlinear damping has a more significant influence on jump points. Secondly, when the values of $V_{AC}$ and $V_{DC}$ exceed the values of the jump point, the nonlinear damping may outstandingly affect the amplitude.

It can be seen from Equation (9) that the coefficient $C_{2}$ in the cubic nonlinear damping term is worthy of particular attention among the three damping terms. From Equation (5), we have
(30)$$C_{2}=\frac{18d^{2}}{h^{2}}C_{1},$$
so $C_{2}=0.045C_{1}$, $c_{2}=1.5c_{1}$, and $c_{3}=3.4c_{1}$. Table 2 gives three sets of damping parameters that will be used in the present paper.

The nonlinear damping terms are smaller than these of Huang [27] because the beam’s length in this paper is greater than that of Huang’s. This may explain why nonlinear damping has little effect on vibrations in this study. According to Equation (30) and Figure 13, both linear damping $c_{1}$ and cubic nonlinear damping $c_{2}$ are positive, which usually decreases vibration amplitude and shifts jump points towards the right. Particularly, the quadratic nonlinear damping $c_{3}$ has an inverse effect that increases the vibration amplitude and makes the jump point shift to the left. As shown in Figure 12 and Figure 13, $c_{2}$ and $c_{3}$ play crucial roles. According to Huang’s conclusion [27] and Equation (30), the effect of nonlinear damping can be enhanced by increasing the distance d or reducing the thickness of the beam h.

In summary, the small-scale effect has a significant impact on the mechanical properties of microbeam resonators. If the small-scale effect is ignored, the load may excite a larger amplitude than that with the small-scale effect. In fact, by neglecting the small-scale effect, the nonlinear damping may lead to wrong results. Therefore, small-scale effect and nonlinear damping must be considered for the accurate description of microbeam resonators’ vibration.

Now, we focus on the effects of the small-scale effect and the nonlinear damping on the first mode. So, the electric-force term in Equations (8) and (9).
(31)$$F=F_{1}+F_{2}\cos\left(\Omega t\right)+F_{3}\cos\left(2\Omega t\right).$$

Here, $F_{1}=2\pi^{-1}Q\left(2V_{DC}{}^{2}+V_{AC}{}^{2}\right),\;F_{2}=8\pi^{-1}QV_{DC}V_{AC},\;F_{3}=2\pi^{-1}QV_{AC}{}^{2}.$ This means that the external excitations depend on the DC voltage $V_{DC}$ and AC voltage $V_{AC}$. $F_{3}$ may be neglected from Equations (28) and (29) due to $V_{DC}\gg V_{AC}$.

As shown in Figure 14, Figure 15 and Figure 16, if other parameters are fixed, the larger $V_{DC}$ and $V_{AC}$ lead to the bigger vibration amplitude. When the exciting frequency reaches above the modal frequency, the external excitation and the nonlinear terms cause the response curve to bend, thus resulting in multi-values of amplitude. This induces the jump of the vibration amplitude at the bifurcation points. 

We can use a surface to show the combined effects of $V_{DC}$ and $V_{AC}$ on the vibrational amplitude, as shown in Figure 17 and Figure 18. The two figures indicate two main conclusions. Firstly, the effect of $V_{DC}$ and $V_{AC}$ have about equal influence on the amplitude of vibration. Secondly, when the load’s frequency exceeds the modal frequency, the jump will occur in the primary resonance. 

The magnitude of the load is determined not only by the DC and AC voltages but also by the material parameter $Q=\varepsilon_{0}b/2md^{3}\omega_{0}^{2}$. The sensitivity of the resonator (a small external load produces a big amplitude) can be improved by increasing the length of the microbeam. Moreover, reducing the distance *d* is an effective way to improve the sensitivity of the device. For example, for the given $V_{AC}$, a smaller *d* will lead to a greater amplitude of vibration, as shown in Figure 19.

## 6. Conclusions

In the present paper, we propose a new partial differential equation to model nonlinear oscillations of the microbeam resonator with the electric force. This model includes the small-scale effect and the nonlinear damping terms. We obtain nonlinear ordinary differential equations for first and third modes by the Galerkin method. Then, their approximate analytical solutions are obtained by the multiple-scale method for the primary resonance. And the solutions are used to study the influences of the driving electric field, small-scale effect, and nonlinear damping on the vibrations of the structure. The results suggest the following:

(1) Under the same excitation voltage, the amplitudes of the third mode are much smaller than these of the first mode.

(2) The small-scale effect has a significant impact on both static loading and dynamic vibration. The nonlinear damping has a small effect on the vibration amplitude when the load’s frequency is less than the modal frequency. However, when the load’s frequency is greater than the modal frequency, nonlinear damping will change the jump points of the load. 

(3) Both $V_{DC}$ and $V_{AC}$ have a significant effect on the vibration amplitude for the primary resonance. When the exciting frequency is greater than the modal frequency of the microbeam, the external excitation terms and the nonlinear damping terms cause the response curve to bend and result in the multi-value amplitude and jumping phenomena.

## Figures and Tables

**Figure 1 micromachines-14-00170-f001:**
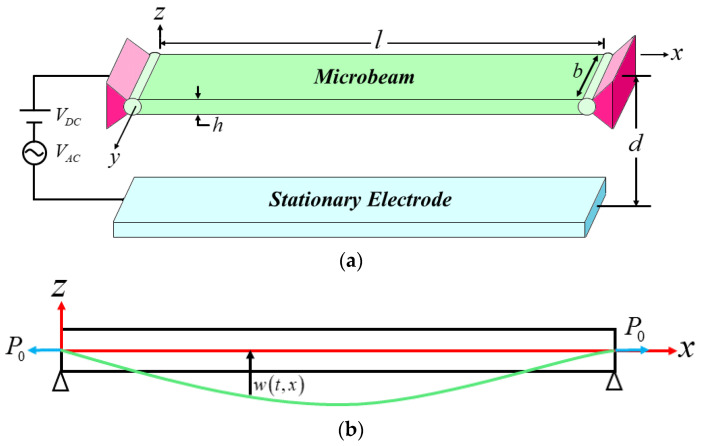
Microresonator model: (**a**) Schematic drawing of the resonator; (**b**) structural characteristic drawing of microbeam.

**Figure 2 micromachines-14-00170-f002:**
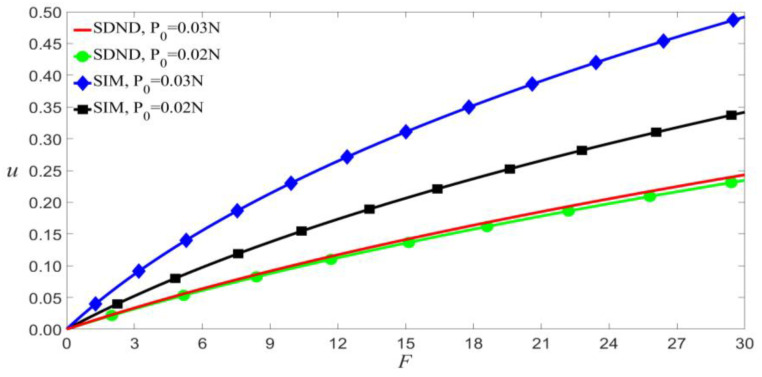
Impact of small-scale effect and initial axial load on the static bending of the microbeam for $V_{DC}=1.5\;V$ and $V_{AC}=0.08\;V$.

**Figure 3 micromachines-14-00170-f003:**
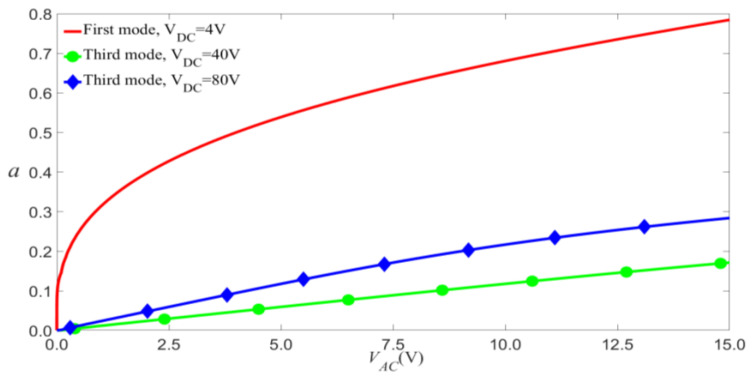
Amplitude of the response as a function of the load’s amplitude of primary resonance for σ=0.

**Figure 4 micromachines-14-00170-f004:**
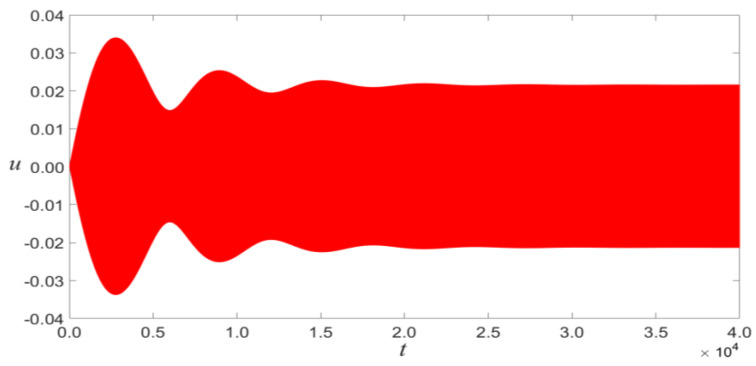
Time evolution of the midpoint displacement of microbeam under the context of primary resonance for $V_{DC}=1.5\;V$, $V_{AC}=0.08\;V$, and σ=0.1.

**Figure 5 micromachines-14-00170-f005:**
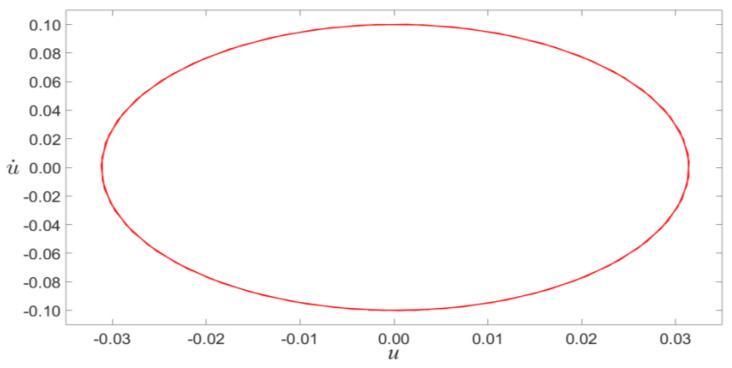
The phase portraits of primary resonant excitation for $V_{DC}=1.5\;V$, $V_{AC}=0.08\;V$, and σ=0.1.

**Figure 6 micromachines-14-00170-f006:**
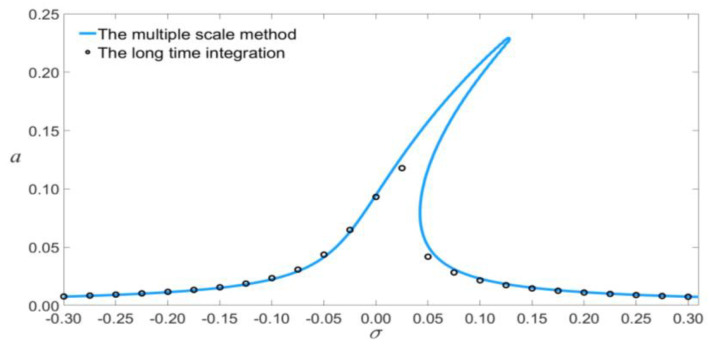
Comparison between multiple-scale method and the Runge–Kutta method.

**Figure 7 micromachines-14-00170-f007:**
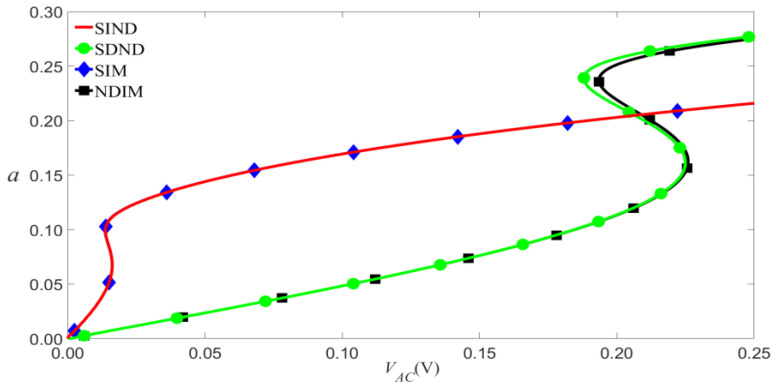
Amplitude of response as a function of the amplitudes of load’s for microbeam for $V_{DC}=4\;V$, $P_{0}=0.01\;N$, and σ=0.05.

**Figure 8 micromachines-14-00170-f008:**
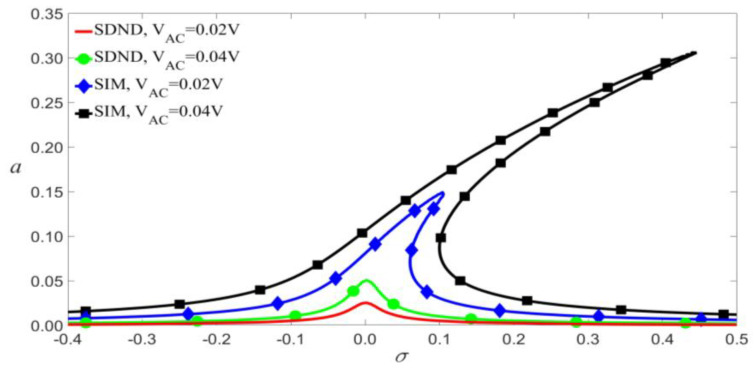
Frequency-response curves of primary resonance for $V_{DC}=4\;V$ and $P_{0}=0.01\;N$.

**Figure 9 micromachines-14-00170-f009:**
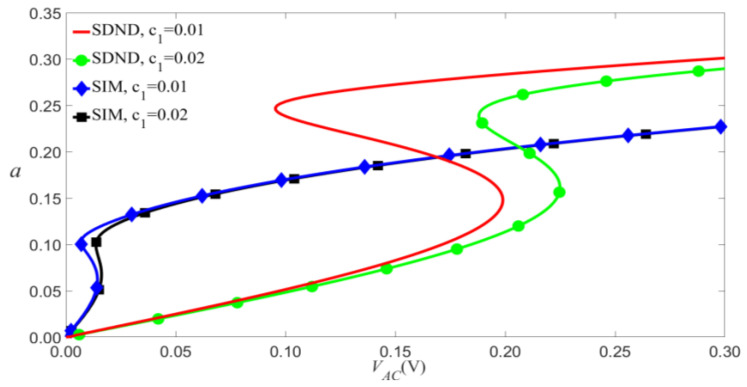
Amplitude of the response as a function of the load’s amplitude of primary resonance for $V_{DC}=4\;V$, $P_{0}=0.01\;N$, and σ=0.05.

**Figure 10 micromachines-14-00170-f010:**
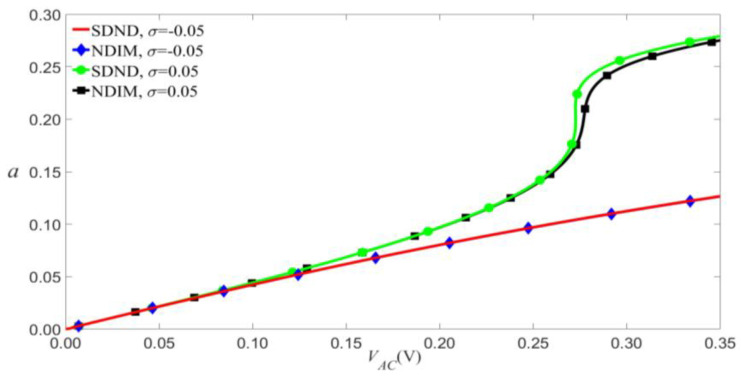
Amplitude of the response as a function of the load’s amplitude of primary resonance for $V_{DC}=4\;V$, $P_{0}=0.01\;N$, and $c_{2}=0.03$.

**Figure 11 micromachines-14-00170-f011:**
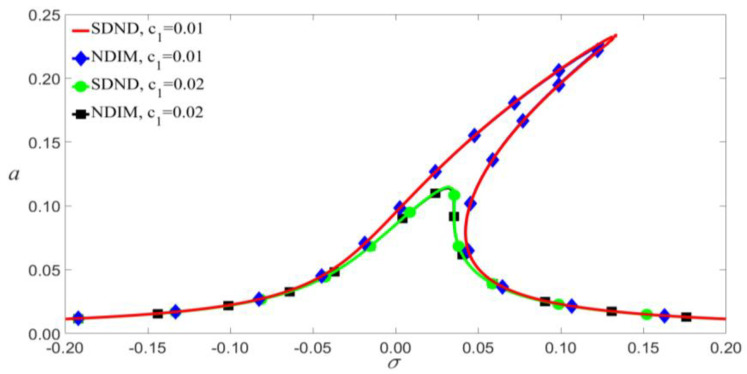
Frequency-response curves of primary resonance for $V_{DC}=1.5\;V$, $V_{AC}=0.08\;V$, and $P_{0}=0.205\;N$.

**Figure 12 micromachines-14-00170-f012:**
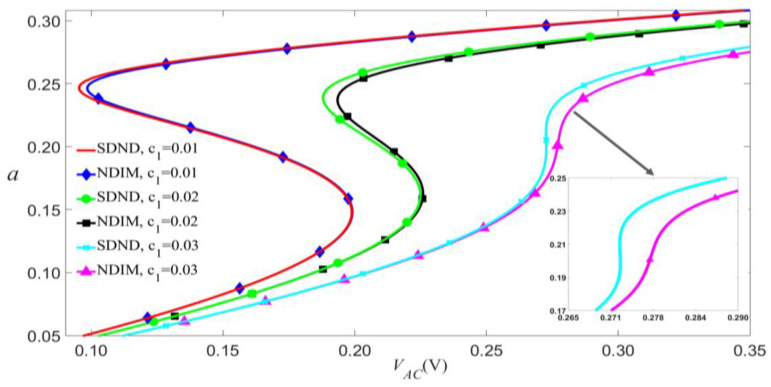
Amplitude of the response as a function of the load’s amplitude of primary resonance for $V_{DC}=4\;V$, $P_{0}=0.01\;N$, and σ=0.05 given several damping parameters.

**Figure 13 micromachines-14-00170-f013:**
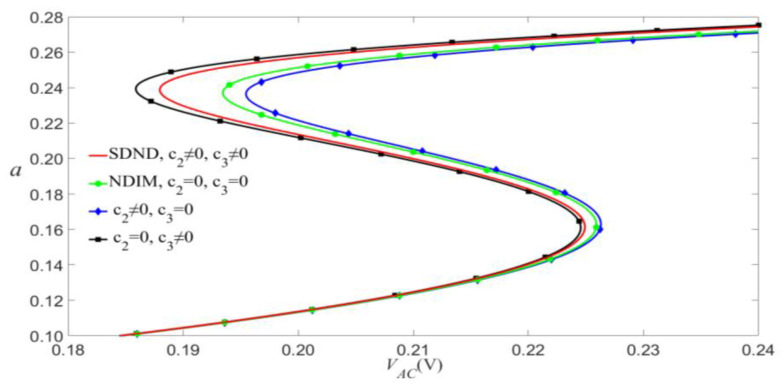
Comparison in amplitude of the response as a function of the load’s amplitude by considering different nonlinear damping terms.

**Figure 14 micromachines-14-00170-f014:**
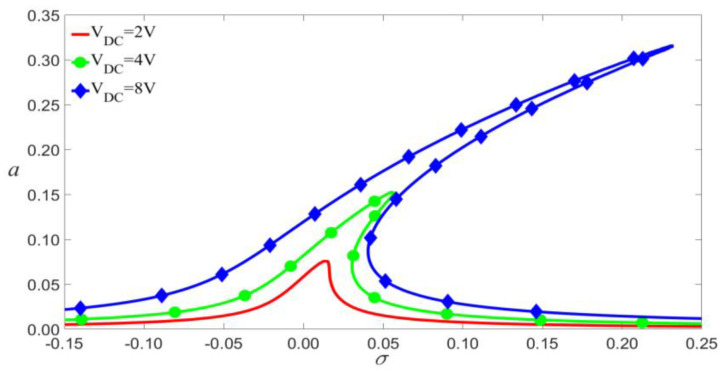
Frequency-response curves of primary resonance for $V_{AC}=0.02\;V$.

**Figure 15 micromachines-14-00170-f015:**
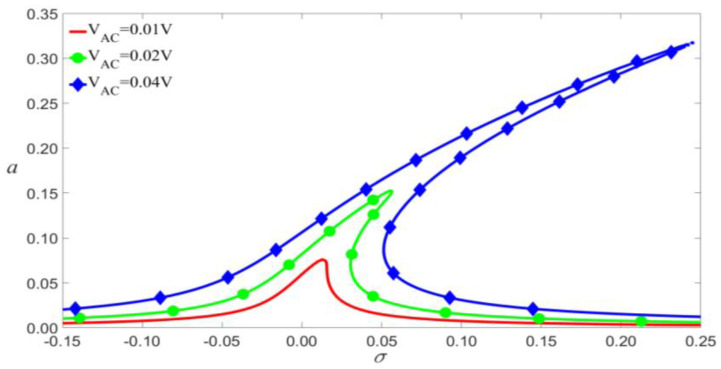
Frequency-response curves of primary resonance for $V_{DC}=4\;V$.

**Figure 16 micromachines-14-00170-f016:**
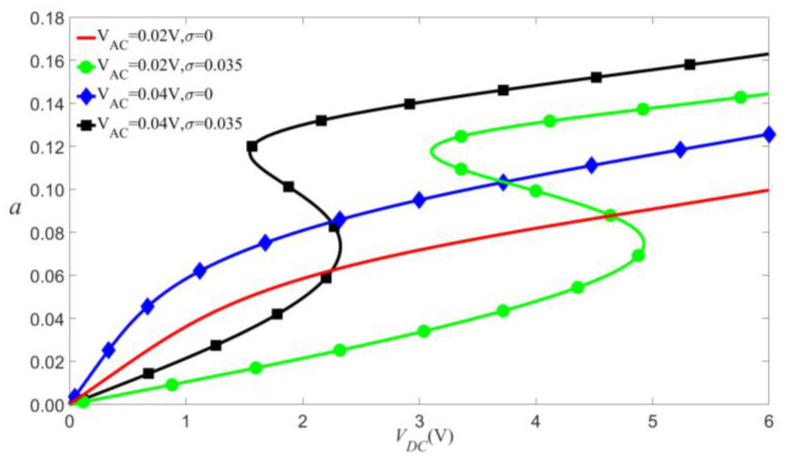
Amplitude of the response as a function of the load’s amplitude of primary resonance.

**Figure 17 micromachines-14-00170-f017:**
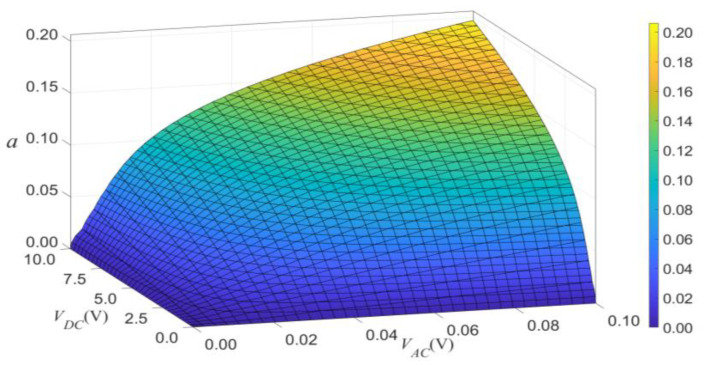
The 3D plot of amplitude of the response as a function of the load’s amplitude of primary resonance for σ=0.

**Figure 18 micromachines-14-00170-f018:**
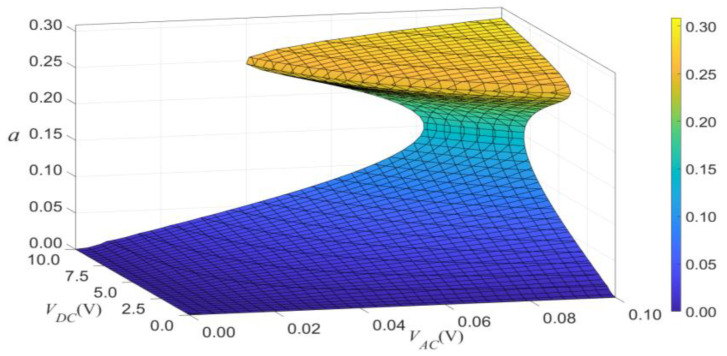
The 3D plot of amplitude of the response as a function of the load’s amplitude of primary resonance for σ=0.05.

**Figure 19 micromachines-14-00170-f019:**
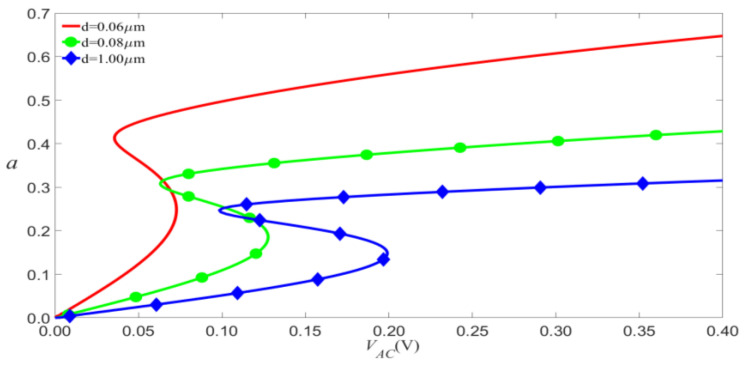
Amplitude of the response as a function of the load’s amplitude of primary resonance with $V_{DC}=4\;V$ and σ=0.05 for d.

**Table 1 micromachines-14-00170-t001:** Geometric and material parameters of MEMS resonator [27,31].

Quantity	Values	Quantity	Values
Length, *L*	500 (µm)	*S = bh*	200 (µm^2^)
Thickness, *h*	20 (µm)	*I = (bh^3^)/12*	6.670 × 10^3^ (µm^4^)
Width, *b*	10 (µm)	*m = ρS*	4.600 × 10^−7^ (kg/m)
Young’s modulus, *E*	165 (Gpa)	*G*	58.750 (Gpa)
Density, *ρ*	2300 (kg/m^−3^)	Initial axial load, *P_0_*	0.205 (N)
Capacitor gap width, *d*	1 (µm)	*c* _1_	1.000 × 10^−2^
The dielectric constant, $\varepsilon_{0}$	8.85 × 10^−12^ (F/m)	*c* _2_	1.568 × 10^−4^
Material length scale parameter, *ξ*	20 (µm)	*c* _3_	−3.395 × 10^−4^

**Table 2 micromachines-14-00170-t002:** Damping parameters [27].

*c* _1_	*c* _2_	*c* _3_
1.000 × 10^−2^	1.568 × 10^−4^	−3.395 × 10^−4^
2.000 × 10^−2^	3.135 × 10^−4^	−6.791 × 10^−4^
3.000 × 10^−2^	4.703 × 10^−4^	−1.000 × 10^−3^

## Data Availability

The data presented in this study are available in the article insert.

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
