# Peer review of "Impacts of Small-Scale Effect and Nonlinear Damping on the Nonlinear Vibrations of Electrostatic Microresonators"

_micromachines, 2023, doi:10.3390/mi14010170_

Round 1

Reviewer 1 Report

The authors present an investigation of small-scale effect and nonlinear damping on the nonlinear vibrations of electrostatic microresonators. However, it is found that the present nonlinear model has been well developed according to the authors’ previous publications. The governing equation based on the Euler-Bernoulli beam theory was built and discussed in detail in the ref. [26], and the nonlinear damping effect and small-scale effect were also investigated in the reference. The multiple scale method is also a widely used method for solving nonlinear equations. The only difference is that the electrostatic force is involved in this work, but there is no more innovation point in this work. To read this paper, authors have to find a lot of formulas and parameter definitions from the previous reference. This is not easy to follow the mathematical model.

Moreover, the material that used for microresonator is not clear and only the properties are given. Authors must confirm the Kelvin-Viogt amping model can be used for this microresonator or not. Authors must consider real material types and apply real damping models to increase the physical significance of the study.

Author Response

Thank you for your letter dated December 28. We thank the reviewers for the time and effort that they have put into reviewing the previous version of the manuscript. Your suggestions have enabled us to improve our work. Based on the instructions provided in your letter, we uploaded the file of the revised manuscript. The following content is our point-by-point response to the comments raised by the reviewers.

Major Remarks:

The authors present an investigation of small-scale effect and nonlinear damping on the nonlinear vibrations of electrostatic microresonators. However, it is found that the present nonlinear model has been well developed according to the authors’ previous publications. The governing equation based on the Euler-Bernoulli beam theory was built and discussed in detail in the ref. [26], and the nonlinear damping effect and small-scale effect were also investigated in the reference. The multiple scale method is also a widely used method for solving nonlinear equations. The only difference is that the electrostatic force is involved in this work, but there is no more innovation point in this work. To read this paper, authors have to find a lot of formulas and parameter definitions from the previous reference. This is not easy to follow the mathematical model.

Moreover, the material that used for microresonator is not clear and only the properties are given. Authors must confirm the Kelvin-Viogt amping model can be used for this microresonator or not. Authors must consider real material types and apply real damping models to increase the physical significance of the study.

Reply:Although we use the microbeam’s model in Ref. [27], our model have two new crucial features induced by the loads.

One is that when the denominator of the term of electricity excitation is multiplied to the left of Eq. (4) and truncates the partial differential equation using the Galerkin method, some nonlinear terms with the scale effect appear in Eqs. (8). This can be seen in Eqs. (9) and (10). These coupling terms have a remarkable influence on the structure’s vibrations, as shown in Figs. 3, 8, and 14.

Second, when the influence of the voltage is considered, the constant terms (F1j) will appear in Eqs. (8). Solving a nonlinear ordinary differential equation with constant terms has not been profoundly studied so far. In this paper, we solve, for the first time, the equation with constant term using multiscale methods. The solutions show that the multiscale method is still valid for the equation, as shown in Fig.6. Our research provides an example for solving nonlinear differential equations with constant terms.

We used the parameters in the manuscript from Refs. [27]. These parameters are frequently used in existing studies. Ref. [31] was added to explain the material of the resonator in the revision. Moreover, the parameters in the manuscript are suitable for demonstrating the effect of voltage on structural dynamics. We add the above interpretation in lines from 155 to 166 of the manuscript in red.

We use Kelvin-Viogt damping model to describe the energy dissipation of microstructures. In fact, it is an open question how describe the microstructure’s energy dissipation. However, recent studies have shown that the Kelvin-Viogt damping model is qualitatively suitable for the vibration of microbeams. For example, reference [26] shows that nonlinear dissipation has a significant impact on the dynamics of micromechanical systems, and the Voigt–Kelvin viscoelastic constitutive law suits to describe the linear and nonlinear damping. We add the comments of damping in lines 113 to 120 in the revision in red.

We again thank you for your patience, tolerance and comments, which significantly improved the quality of the paper. The content of the changes we upload is marked in red. We would like also to thank you for allowing us to resubmit a revised copy of the manuscript. We hope that the revised manuscript is accepted for publication in the Micromachines.

Best wishes

Dayao Meng

Jan. 4, 2023

Reviewer 2 Report

This manuscript proposed a new microbeam model for predicting the small-scale effect and nonlinear damping on the nonlinear vibration of electrostatic resonators, which is useful for engineering applications. The results are reasonable. So, this reviewer recommends to publish in this journal with minor revision. More detailed comments for the manuscript are listed below.

1. Is there a buckling problem in the current model?

2. The quality of Figs. 2, 3, 4, 6, 8, 10, 11, 13, 16, 17 and 19 should be imporved (i.e., delete the extra grey lines below the figure.).

3. The references part could be extended considering some recent/previously related studies such as: Crystals11(10), 1206, 2021.

Author Response

Thank you for your letter dated December 28. We thank the reviewers for the time and effort that they have put into reviewing the previous version of the manuscript. Your suggestions have enabled us to improve our work. Based on the instructions provided in your letter, we uploaded the file of the revised manuscript. The following content is our point-by-point response to the comments raised by the reviewers.

Major Remarks:

Comment 1: Is there a buckling problem in the current model?

Reply: The microresonator does not buckle when . When the initial axial load is within a certain interval, our model will have buckling. Taking your good suggestion, we made modifications in lines 178-183 in red.

Comment 2: The quality of Figs. 2, 3, 4, 6, 8, 10, 11, 13, 16, 17 and 19 should be imporved (i.e., delete the extra grey lines below the figure.).

Reply: Taking your good suggestion, we have reformatted the image to make sure that no extra grey lines below the figure.

Comment 3: The references part could be extended considering some recent/previously related studies such as: Crystals, 11(10), 1206, 2021.

Reply: We read the article you suggested and think that it is well written. The theory and research methods involved in it have helped us a lot, and we add this article you recommended. Add Ref. [9] based on your suggestions. These modify is shown in lines 33-34.

We again thank you for your patience, tolerance and comments, which significantly improved the quality of the paper. The content of the changes we upload is marked in red. We would like also to thank you for allowing us to resubmit a revised copy of the manuscript. We hope that the revised manuscript is accepted for publication in the Micromachines.

Best wishes

Dayao Meng

Jan. 4, 2023

Round 2

Reviewer 1 Report

The paper can be recommended for publication now.